# Temporal and spatial interactions in sympatric ungulates: Insights from Japanese serow and sika deer

Tomoki Mori [1¤*], Kensuke Miura[2], Hiroyuki Takeuchi[2], Yasuaki Niizuma[3]

1 Institute for Mountain Science, Shinshu University, Kamiina County, Nagano Prefecture, Japan, 2 Graduate School of Agriculture, Meijo University, Tenpaku-ku, Nagoya, Japan, 3 Laboratory of Environmental Zoology, Faculty of Agriculture, Meijo University, Tenpaku-ku, Nagoya, Japan

¤ Current address: Research Center for Wildlife Management, Gifu University, Gifu, Japan
* tmkmori12@gmail.com

## Abstract

Sympatric species, commonly evolve behavioural mechanisms allowing them to coexist, thereby reducing direct competition for resources. In Japan, since the 1970s, the endemic Japanese serow (*Capricornis crispus*) and the sika deer (*Cervus nippon*) have been primarily allopatric. However, due to the rapid expansion of the sika deer population on Japan's main island of Honshu, the habitats of these two species now overlap. The significant and increasing overlap raises concerns about the potential impacts between these two (now sympatric) ungulates, including changes in distribution, shifts in activity patterns, or displacement due to interspecific competition. In this study, we investigated temporal and spatial segregation between Japanese serow and sika deer, from 2015 to 2017, in Shirakawa Village, Gifu Prefecture, Japan, by means of camera traps. Although our study was limited by a small sample size, it revealed no clear temporal or spatial segregation between the species, suggesting that there is potential for coexistence in shared habitats without pronounced competitive conflict, perhaps due to an abundance of food relative to sika deer density. Nevertheless, during autumn, reduced activity overlap rates, when the relative abundance index (RAI) of sika deer increased, may indicate that Japanese serow have modified their behavior to minimize resource competition. Specifically, during summer, when the RAI of sika deer was low, Japanese serow exhibited cathemeral behavior, whereas in autumn Japanese serow became nocturnal as sika deer RAI values increased. This seasonal adjustment indicates a context-dependent behavioural response that may serve to reduce temporal overlap and mitigate competition. Given the increasing sika deer population, understanding this potential for intensified competition becomes crucial for effective wildlife management and conservation efforts, particularly in maintaining the ecological balance between these species and ensuring the long-term sustainability of their habitats.

**Data availability statement:** All relevant data are within the paper and its Supporting Information files.

**Funding:** The author(s) received no specific funding for this work.

**Competing interests:** The authors have declared that no competing interests exist.

## Introduction

Sympatric species are those that coexist in the same geographic area, often sharing similar ecological niches, which creates the potential for competition over limited resources such as habitat and food [1]. Such competition could lead to displacement of one species, or even its extinction, if not for certain ecological mechanisms that act to mitigate it [2]. Sympatric species, commonly evolve multiple strategies that promote coexistence, ensuring the persistence of biodiversity and the stability of ecological systems [3]. A key mechanism, facilitating the coexistence of sympatric species, is spatial and temporal partitioning of their activities, thereby reducing the negative effects of direct competition and optimizing resource use. This mechanism is evident across a diverse range of animal species, including herbivores. For example, in Tanzania, the spatial and dietary segregation between zebra (*Equus quagga*) and wildebeest (*Connochaetes taurinus*) allow them to coexist despite occupying overlapping habitats [4]. In Kenya, three sympatric jackal species partition their activity times, with golden jackal (*Canis aureus*) being active during the day, while black-backed jackal (*Lupulella mesomelas*) and side-striped jackal (*L. adusta*) are mostly nocturnal [5], while in USA, mule deer (*Odocoileus hemionus*) and white-tailed deer (*O. virginianus*) show different spatial and habitat use patterns, allowing them to coexist seasonally despite potential competition between them [6].

On Honshu, the main island of Japan, two large ungulates, the endemic Japanese serow (*Capricornis crispus*) (hereafter serow) and the more wide-ranging sika deer (*Cervus nippon*) (hereafter deer), inhabit diverse landscapes, sharing similar ecological characteristics (e.g., they are both herbivorous, and weigh over 30 kg) and form essential components of diverse forest and montane ecosystems. Although serow populations remain stable in some regions, localized declines have led to their designation as "Threatened Local Populations" in the Red Data Book of Japan [7], particularly in Kyushu, Shikoku, the Suzuka Mountains, and the Kii Mountains. The serow, which is well-known for its territorial behavior [8,9], primarily inhabits montane forests, particularly deciduous forests, and has a relatively narrow, selective diet [10]. It also inhibits alpine meadows, although it's diet in these areas is relatively broader [11,12]. In contrast, the deer, which occurs widely throughout the Japanese archipelago, is not territorial, has a broad diet that includes grasses, forbs, and dicotyledonous species [13,14]. Until the 1970s, these species were allopatric, except in a few isolated areas in northern Honshu where their distributions overlapped [15,16]. However, since the 1990s the deer population has increased and its distribution (including elevation) has expanded dramatically [17], driven by factors such as reduced hunting pressure, changes in land use, and milder winters [18]. As a result, deer have steadily increased the area over which they now occur sympatrically with serow [9]. Such an expansion in deer numbers and distribution increases interspecific interactions between deer and serow, especially in habitats with limited food options [19].

Deer can be considered to have a competitive advantage over serow due to their larger body size, social behavior, and broad diet [20–22]. Their ability to exploit a wider range of food resources allows them to outcompete the more specialized serow, particularly in resource-limited habitats [21]. Furthermore, some studies report

that serow avoid deer in areas with high deer density [22]. Although serow occasionally display territoriality behaviors toward deer, they rarely succeed in displacing them [23]. Overall, the combination of a larger body size, higher population density, and greater dietary flexibility is thought likely to provide sika deer with a long-term competitive edge in sympatric habitats.

Interspecific interactions between serow and deer vary according to the local environmental context. For example, in Gunma Prefecture, central Honshu, they exhibit distinct habitat preferences, with serow favoring steep forested slopes and deer opting for forested areas with gentle terrain away from human settlement [24]. Their periods of activity also vary, with serow being diurnal and deer being crepuscular [24]. Further to the south, on Mount Fuji, both species favor broad-leaved forest, and share more than 90% of their habitat due to the uniformity of the terrain and the limited plant diversity, with over 80% activity overlap [25]. Their diets may overlap in low-diversity habitats with scarce food resources [19].

Furthermore, the competitive relationship between these two species may vary depending on seasonal changes in behavior. During the autumn mating season, male deer congregate at mating sites and establish harems of several females each [17], thereby leading to a temporary rise in their local population density. For example, in the Kaga region of Ishikawa Prefecture (on the Sea of Japan coast of Honshu), there was a notable increase in deer observations during autumn (based on camera traps) [26] and in addition heightened activity and aggressiveness of territorial male deer during the autumn rut [17], which we suspect leads to more frequent interaction between deer and serow in autumn compared with summer. In response to these changes, even where deer density is low, serow may adjust their habitat use or period of activity to avoid direct encounters with deer. Such seasonal variation provides an opportunity to study seasonal interaction dynamics between serow and deer.

In order to identify the temporal and spatial activity patterns of serow and deer in summer and autumn, we investigated the activity and space utilization patterns of both species by using camera traps in Shirakawa Village, Gifu Prefecture (in central Honshu), from 2015–2017. Shirakawa Village still has a relatively low deer density, although future increases are anticipated, making it an ideal location to study early-stage interactions before competition intensifies. We hypothesized that even under such low-density conditions, the autumnal increase in local deer activity could alter interaction dynamics between the two species, leading to distinctive patterns of temporal or spatial segregation in their shared habitats. Our findings contribute to a better understanding of such interspecific interactions while providing insights for conservation and management.

## Materials and methods

### Study area

Our study was conducted in Shirakawa Village (36°16′18″N, 136°54′23″E), in northwestern Gifu Prefecture, Japan (Fig 1), at the foot of Hakusan (a mountain rising to 2,700 m), within an elevation range of 600–1,600 m, covering an area of approximately 130 km². It is characterized by steep, rugged terrain with deep valleys and is predominantly covered with broad-leaved forests, with approximately 12% consisting of patchily distributed Japanese cedar (*Cryptomeria japonica*) plantations at elevations below 1,600 m [27]. The local area has a humid continental climate, with warm, humid summers (averaging 21.8°C from 2015–2017), and cold, snowy winters (averaging 0.9°C over the same period).

The forest vegetation below 800 m consists of broad-leaved trees, such as konara oak (*Quercus serrata*) and Japanese chestnut (*Castanea crenata*). Above 800 m but below 1,600 m, Mongolian oak (*Q. crispula*) and Japanese beech (*Fagus crenata*) predominate. The shrub-layer decline rank evaluation [SDR, 28] in 2016 indicated minimal deer-induced vegetation decline in the study area [29], indicating no observable degradation of vegetation in the study area during the study, and food resources for both serow and deer were considered to be abundant (Fig 1).

Human disturbance (of both serow and deer) in the study area is considered minimal for the following reasons. First, all survey sites were at least 600 m from human settlements, reducing direct human influence. Ikeda et al. [30] found that deer are more common in areas with low human disturbance, using a 500 m buffer, suggesting our sites experience

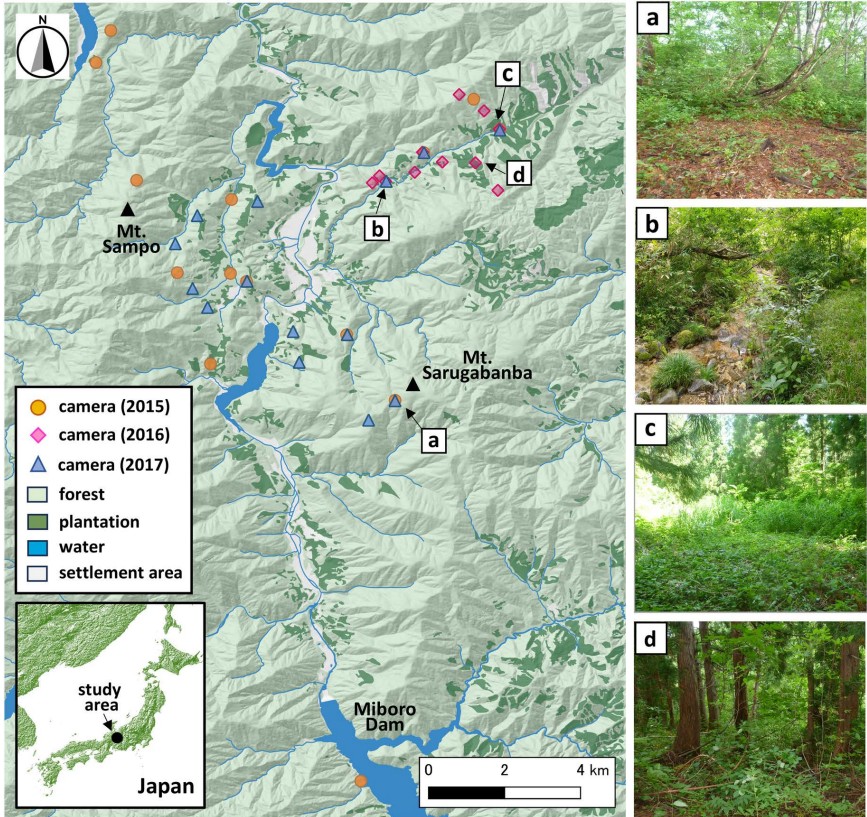

**Fig 1. Locations of 39 camera traps and representative habitats in Shirakawa Village, Gifu Prefecture, Japan (2015–2017).** Camera traps were set in 2015 (orange circles), 2016 (pink diamonds), and 2017 (blue triangles). The right panel shows typical habitats at camera trap sites: (a) deciduous forest, (b/c) riparian forest, (d) coniferous plantation. We created this map ourselves using data provided by the Geospatial Information Authority of Japan (https://www.gsi.go.jp/) and Biodiversity Center of Japan (http://gis.biodic.go.jp/webgis/), under the CC BY 4.0 or a compatible license.

limited impact. Second, in the case of 2017, no deer were captured during our survey period (excluding the hunting season), and only a few were captured outside this period [29], further minimizing human influence. Third, serow are generally less sensitive to human activity than deer [30] and have been protected as a Special Natural Monument since 1955, with strict capture regulations.

Various mammal species (in addition to deer and serow) occur in the various habitats within the study area. These include: Asiatic black bear (*Ursus thibetanus*), wild boar (*Sus scrofa*), raccoon dog (*Nyctereutes procyonoides*), red fox (*Vulpes vulpes japonica*), and Japanese hare (*Lepus brachyurus*) [27]. In recent years, the deer population has been increasing [29].

## Camera trapping

Over a period of three years (2015–2017), we installed 39 camera traps (14 in 2015, 11 in 2016, and 14 in 2017) in various habitats at various elevations, to ensure a comprehensive representation of the local environment. Camera traps were installed from 1 July to 31 October each year. The study period was set after the birthing season (mainly May to June: [17]), minimizing population fluctuations due to new births. While deer are known to exhibit seasonal movements in regions with heavy snowfall [31,32], the study was conducted before snowfall, reducing the likelihood of such movements. Therefore, demographic closure was likely maintained during the study period.

In 2015, we installed 14 cameras (13 FieldnoteDUO, Fieldnote; Marif, Yamaguchi, Japan, and one Reconyx XR6, Reconyx Inc., Holmen, WI, USA). In 2016, we installed 11 cameras (FieldnoteDUO), and in 2017 we installed 14 cameras (nine FieldnoteDUO and five Hyke cam SP2, Hyke inc., Asahikawa, Japan). Although the placement of camera traps varied annually, certain locations were utilized consistently, with cameras positioned at nearly identical coordinates (Fig 1). Camera traps were installed based on the survey routes established by Mori et al. [33] and their surroundings, as well as other key locations, proportionally distributed across vegetation types: broad-leaved forest, plantations, and riparian forest located along streams and rivers. The average inter-camera distance was 1,028±602 m, though terrain constraints resulted in some placements as close as ~350 m. Given the home range sizes of serow (0.16–1.62 km$^2$: [10]) and deer (0.19–2.59 km$^2$: [32,34]), independence was largely ensured, though some individuals may have been recorded by multiple cameras. To capture a wide range of environmental conditions, cameras were also distributed across different topographic features, including valleys, ridges, and slopes. These vegetation and topographic variations were considered to assess potential differences in habitat use by the two focal species. The average slope at camera trap locations was 15.1° (range 2.2–37.1°). Cameras were mounted at approximately 1.5 m above ground level on trees, with the camera angled downwards by 10–15°. Camera traps were visited every two weeks for maintenance, including replacing batteries and SD cards.

The relative abundance index (RAI) for each species was calculated using the following formula (summer 1 July to 31 August; autumn 1 September to 31 October).

$$RAI = \frac{total\ number\ of\ independent\ events}{the\ total\ number\ of\ camera\ trap\ days\ for\ all\ cameras} \times 100$$

In our study, we defined independent events in camera trap surveys based on the 30-minute rule proposed by O'Brien et al. [35].

## Temporal segregation

We examined the temporal segregation of serow and deer by using circular statistics. First, the activity pattern of each species for each season was visualized using kernel density plots and the "overlap" package [36]. Second, the overlap coefficient (Dhat$_4$), spanning from Δ0 (absence of overlap) to Δ1 (total overlap), was calculated [37]. Additionally, the "circular" package [38] in R.4.0.3 [39] was employed to evaluate the statistical significance of differences in activity patterns via Watson's two-sample test.

## Occupancy model approach

We employed occupancy models [40] to estimate occupancy probability ($\psi$) and detection probability ($p$), and to analyze co-occurrence patterns between deer and serow. These models use detection histories from repeated surveys to estimate occupancy probability ($\psi$) and detection probability ($p$) and can incorporate covariates to assess their effects [40]. Occupancy models do not directly determine habitat characteristics but estimate the probability of a species occupying a given site while accounting for imperfect detection.

In our study, we conducted six sampling sessions, each lasting 10 consecutive days, covering the early (days 1–10), middle (days 11–20), and late (days 21–30) periods of each summer and autumn month (July–October). To ensure consistency across years, all months were standardized to 30 days by excluding the 31st day. Data were pooled across three years (2015–2017), resulting in 39 camera trap locations.

## Single-season single species occupancy models

Firstly, we built single-season single species occupancy models [40] using the "unmarked" package [41] in R [39] to determine the habitat characteristics of each species. Detection histories were coded as "1" (presence: captured on camera at

least once), "0" (absence: not recorded), and "–" (missing: camera traps were either not installed or malfunctioned). The camera spacing generally accounted for home ranges, but some were placed closer together, potentially exceeding the species' ranges. Therefore, we followed Mackenzie et al. [40] in interpreting "occupancy" as "habitat use".

We pooled data across years and incorporated "year" as a covariate in a single-season occupancy model, which is appropriate when different locations are sampled each year within a consistent study area [42]. This approach allowed us to estimate annual variation in occupancy while maintaining sufficient sample size. Although a multi-season model [43] could have been used, we did not aim to estimate colonization and extinction, making the single-season model with year as a covariate a more suitable choice. Four covariates were used to assess the influence of occupancy probability of each species: dominant habitat type (BF broad-leaved forest; CF conifer forest, and RF riparian forest), year (2015, 2016 and 2017), and slope gradients and altitude (continuous covariate). The values for slope gradients and altitude were obtained via 10 m-resolution Digital Elevation Model (DEM) data from the Geospatial Information Authority of Japan.

We conducted model selection by treating both occupancy and detection probabilities as functions of environmental and temporal covariates. Specifically, we used "year" (categorical: 2015, 2016, 2017), "habitat" (categorical: BF, CF, RF), altitude, and slope as explanatory variables for both parameters. The continuous variables, altitude and slope, were standardized prior to analysis to improve model convergence. To prevent overfitting due to the small sample size (n = 39), we constructed models in which both occupancy and detection probabilities included at most one categorical variable (either year or habitat type) and one continuous variable (i.e., slope or altitude). All camera traps were installed based on a standardized protocol. Placement, height, angle, and deployment periods were consistent across sites and years, and the overall detection range did not vary substantially among cameras. Vegetation in front of each camera was cleared as necessary to maintain visibility. These standardized procedures would minimized variation in detection probability due to methodological differences. We used Akaike's Information Criterion adjusted for small sample size (AICc) to rank models [44], and considered models with ΔAICc < 2 as competitive. We estimated covariate effects and by applying model averaging to the competitive models.

**Single-season, two species occupancy model to analyze spatial segregation**

We used $\psi Ba/rBa$ parameterization of single-season, two-species occupancy models [40,45] to investigate co-occurrence patterns for serow and deer using PRESENCE software [46]. This model estimates the occupancy parameters for each species and the conditional probability of occupancy when another species is present or detected [40]. Through this parameterization process, we estimated the following parameters: $\Psi A$ = probability of occupancy for species A; $\Psi BA$ = probability of occupancy for species B, given species A is present; $\Psi Ba$ = probability of occupancy for species B, given species A is absent; $pA$ = probability of detection for species A, given species B is absent; $pB$ = probability of detection for species B, given species A is absent; $rA$ = probability of detection for species A, given both species are present; $rBA$ = probability of detection for species B, given both species are present and species A is detected; $rBa$ = probability of detection for species B, given both species are present and species A is not detected.

We employed two scenarios proposed by MacKenzie et al. [40] for our examination: first, the occupancy of the subordinate species B (serow) being influenced by the existence of the dominant species A (deer) ($\psi BA \neq \psi Ba$) or not ($\psi BA = \psi Ba$). Second, the detection of the subordinate species being influenced by the existence of the dominant species ($pB \neq rBA \neq rBa$) or not ($pB = rBA = rBa$). We incorporated the covariates selected from the single-species occupancy models and performed model selection, followed by model averaging when multiple models with ΔAICc < 2 were selected [44]. As prior research has clearly shown that deer are dominant over serow [47], we did not evaluate the impact of serow on deer.

We also calculated the social interaction factor (SIF) [45] from the best models as follows:

$$ SIF\left(\varphi\right) = \frac{\psi A \psi BA}{\psi A(\psi A \psi BA + (1 - \psi A)\psi Ba)} $$

The estimate of SIF (φ) < 1 suggests that the two species co-occur less frequently than expected under the assumption of independence (i.e., spatial segregation). Conversely, SIF (φ) > 1 indicates that the two species co-occur more frequently than expected under independence (i.e., spatial overlap). A value of SIF (φ) = 1 implies that the two species occupy sites independently.

## Results

During the three years of the study, we recorded 669 summer trap nights in 2015, 574 in 2016, and 675 in 2017, and 796 autumn trap nights in 2015, 600 in 2016, and 596 in 2017, in order to calculate RAI values for both serow and deer. Serow RAI values increased consistently over the three years, with a tendency for higher RAI values during summer. Deer RAI values also increased over the three years, but showed a tendency for higher RAI values during autumn (Fig 2).

### Temporal activity overlaps

Serow exhibited cathemeral activity in summer, with multiple peaks centered around dawn. In autumn, their activity shifted to a predominantly nocturnal pattern (Fig 3). In contrast, deer were crepuscular, showing activity peaks during twilight hours in both summer and autumn. However, their activity tended towards dusk in summer and towards dawn in autumn (Fig 3). Additionally, deer displayed a relatively high level of daytime activity.

Although the temporal activity overlaps between deer and serow were $0.855\Delta_4$ during summer and $0.754\ \Delta_4$ during autumn (Fig 4), they exhibited distinct differences in their peak activity times in autumn (Watson U2 test = 0.338, $P < 0.01$) but not in summer (Watson U2 test = 0.08, $P > 0.1$).

### Single season, single-species occupancy model

For serow, multiple competing models (ΔAICc < 2) were selected for both summer and autumn (Table 1), so model averaging was performed. In summer, occupancy probability tended to be higher in 2016 and 2017 than in 2015 ($\beta \pm SE = 0.33 \pm 0.83$ and $1.33 \pm 2.00$, respectively), but these effects were not significant ($P > 0.1$) (Table 2). Detection probability was negatively affected by altitude, with a significant effect ($\beta \pm SE = -0.37 \pm 0.16$, $P < 0.05$). In autumn, the averaged model included "year" and "altitude" for occupancy and "altitude" for detection, but none of the covariates showed significant effects ($P > 0.1$). For deer, the best model in summer included "year" and "altitude" for detection, with occupancy held constant (Table 1). In autumn, two competing models were selected, and model averaging was applied. In summer, detection probability increased significantly in 2016 and 2017 compared to 2015 ($\beta \pm SE = 1.74 \pm 0.63$, $P < 0.01$; $3.21 \pm 0.64$, $P < 0.01$) and was positively associated with altitude ($0.32 \pm 0.15$, $P < 0.05$) (Table 2). In autumn, occupancy was positively but not significantly associated with altitude ($\beta \pm SE = 0.66 \pm 0.62$, $P = 0.29$), while detection was significantly higher in 2016 and 2017 than in 2015 ($\beta \pm SE = 1.80 \pm 0.49$, $P < 0.01$; $1.48 \pm 0.53$, $P < 0.01$).

Based on these results, we included "year" and "altitude" as covariates for detection probability in the two-species model, while occupancy was modeled without covariates.

### Two species occupancy model

In summer, a single best model was selected based on AICc, with no competing models (ΔAICc < 2). The top two-species model (Akaike weight: 0.60) for serow and deer included year as covariates for detection probability, while occupancy probability was held constant. The model showed that occupancy by deer did not affect the presence of serow ($\psi A = \psi BA = \psi Ba$, SIF = $1.0 \pm 0.0$), and both species maintained relatively high and stable occupancy throughout the study period (Tables 3 and 4). Detection probabilities increased across years for both species (Table 4), indicating a temporal effect on detection. The detection probability of serow was slightly lower in the presence of deer ($pB \neq rBA \neq rBa$), but the difference was minimal.

 

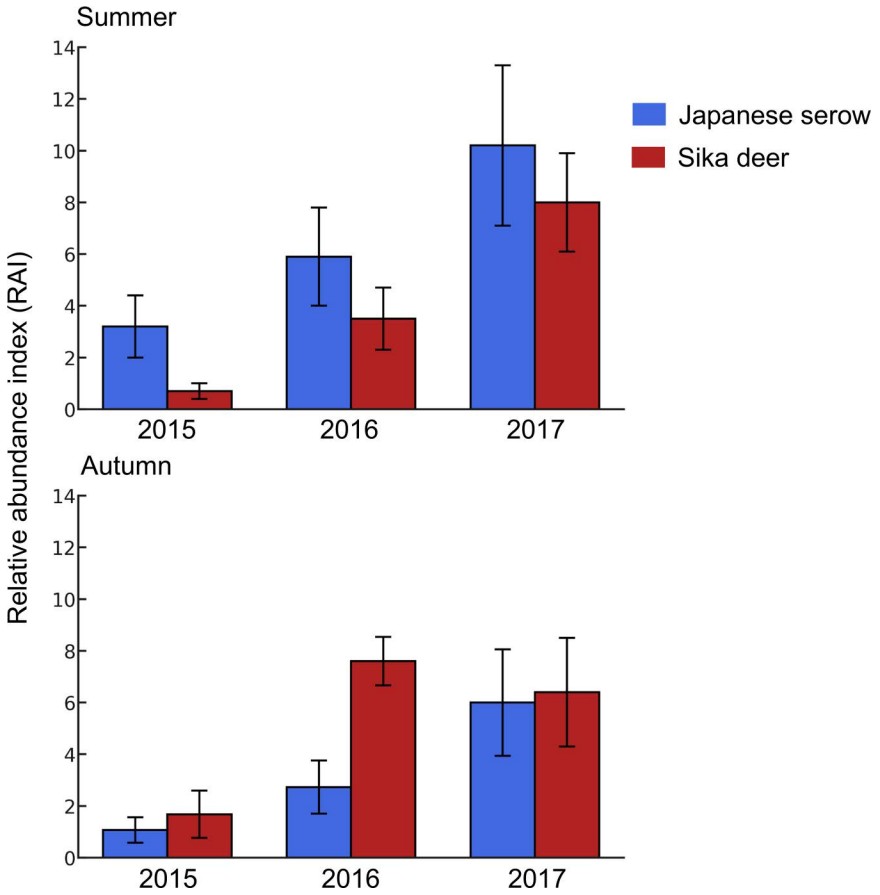

Fig 2. **Seasonal variation in relative abundance index (RAI) and standard error (SE) of Japanese serow and sika deer in Shirakawa Village, Gifu Prefecture, Japan (2015–2017).**

In autumn, a single best model was selected and the top two-species model (Akaike weight: 0.46) included year as a covariate for detection probability and assumed constant occupancy for both species. As in summer, the occupancy of serow was not affected by the presence of deer ($\psi A = \psi BA = \psi Ba$, SIF = 1.0 ± 0.0) (Table 4). Detection probabilities increased over the years for both species. The detection probability of serow remained constant regardless of the presence or absence of deer ($pB = rBA = rBa$).

## Discussion

We investigated the temporal patterns and spatiotemporal segregation of serow and deer using camera traps. Our study was conducted within a limited spatial range, and the relatively small sample size (n = 39) poses a limitation that must be acknowledged. While efforts were made to mitigate potential bias, the results should be interpreted with caution when applied to broader contexts. Additionally, due to our small sample size, we combined data from three years and applied a single-season occupancy model, incorporating "year" as a covariate to account for annual variation while maintaining statistical power. This approach assumes relatively stable occupancy and detection processes across years, unlike a multi-season model, which explicitly estimates colonization and extinction rates. Given our focus on overall interspecific interactions rather than temporal changes in occupancy, this method was the most suitable for our study. Future research using multi-season models may provide deeper insights into long-term dynamics.

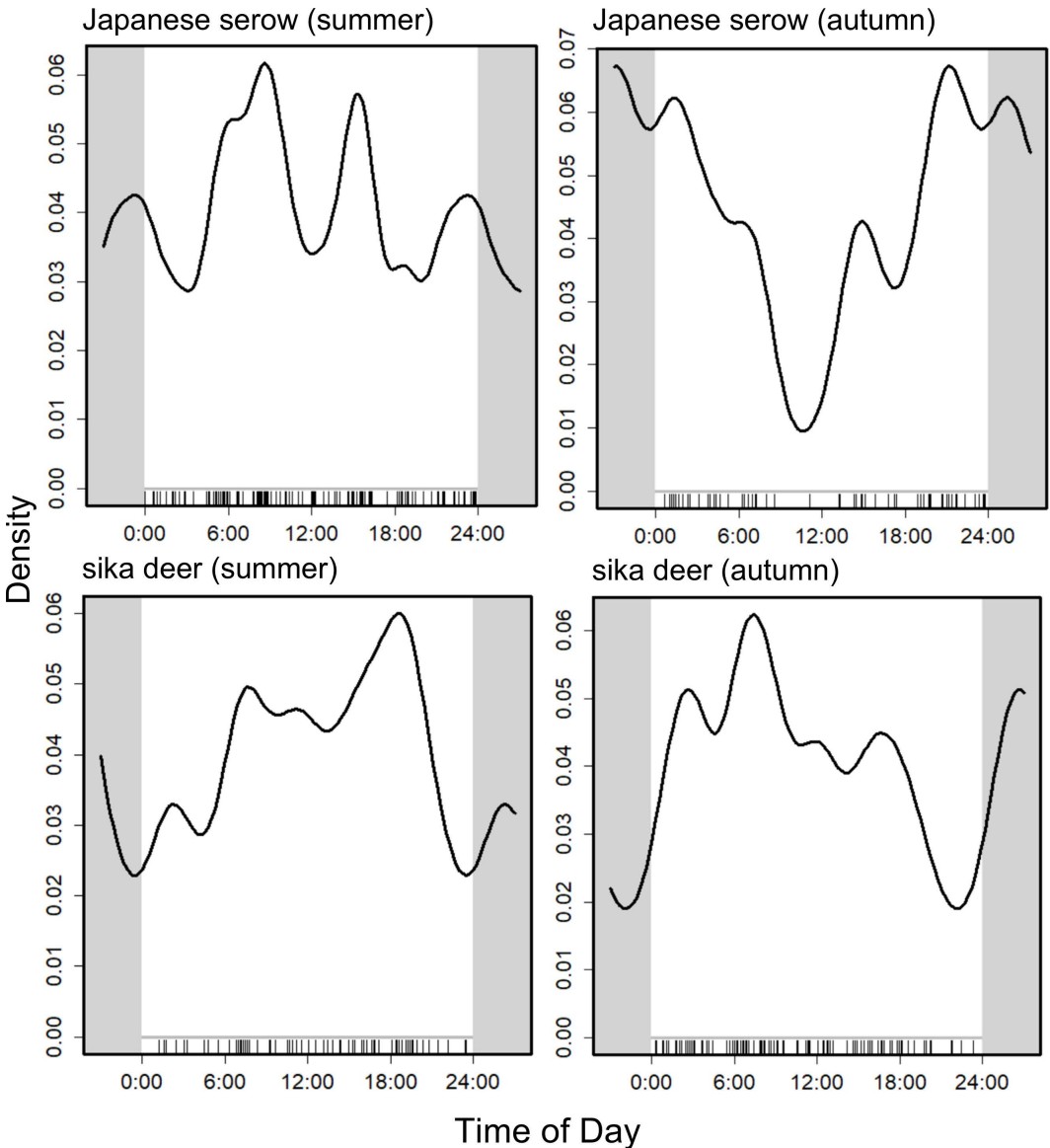

**Fig 3. Activity patterns of Japanese serow and sika deer, across seasons in Shirakawa Village, Gifu Prefecture, Japan (2015–2017).**

Our study showed an annual increase in serow RAI as well as detection probability (as in deer). This increasing trend was consistent with the broader trend of population growth observed across Gifu Prefecture since 2010 [48]. The reasons for the increase in the serow population in Shirakawa Village remain unclear, but this trend might serve as supporting evidence that the species has not yet been affected significantly by the rising deer density.

Our results showed no clear significant impact of deer presence on serow spatial use during the study period, indicating that they might coexist in the same habitats without significant competitive interaction. However, there was a slight reduction in the overlap of activity times in autumn when the RAI of deer increased. While seasonal shifts in food availability could influence activity patterns, previous research in Gifu Prefecture indicates that serow and deer do not exhibit seasonal changes in diel activity [49], suggesting that food availability is unlikely to be the primary factor. Additionally,

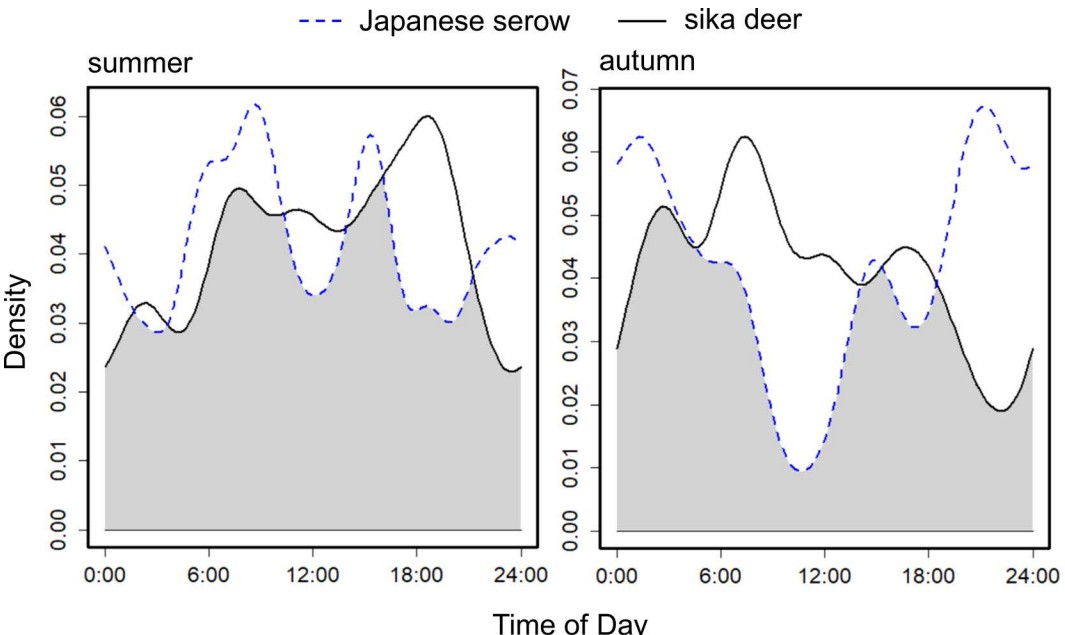

**Fig 4. Overlap plots of the activity pattens of Japanese serow and sika deer, across seasons in Shirakawa Village, Gifu Prefecture, Japan (2015–2017).**

**Table 1. Best-supported single-season occupancy models (ΔAICc<2) for sika deer and Japanese serow in summer and autumn in Shirakawa Village, Gifu Prefecture, Japan. Covariates include year, habitat, slope, and altitude. Ψ=occupancy, p=detection probability, AICc wgt=Akaike weight.**

| Species | Season | Model | AICc | ΔAICc | AICc wgt | no.Par. |
|---|---|---|---|---|---|---|
| Japanese serow | summer | Ψ(.),p(Altitude) | 234.2 | 0.00 | 0.31 | 3 |
| | | Ψ(Year),p(Altitude) | 234.5 | 0.24 | 0.27 | 5 |
| | autumn | Ψ(Year),p(.) | 187.7 | 0.00 | 0.21 | 4 |
| | | Ψ(Year),p(Altitude) | 188.52 | 0.82 | 0.14 | 5 |
| | | Ψ(Year+Altitude),p(.) | 189.04 | 1.34 | 0.11 | 5 |
| | | Ψ(.),p(.) | 189.26 | 1.56 | 0.10 | 2 |
| Sika deer | summer | Ψ(.),p(Year+Altitude) | 191.79 | 0.00 | 0.47 | 5 |
| | autumn | Ψ(.),p(Year) | 214.26 | 0.00 | 0.36 | 4 |

human disturbance is expected to be minimal in our study area. Therefore, serow might be modifying its activity pattern in response to the increasing presence of deer within its range.

## Temporal activity patterns

Numerous mammalian studies have indicated that human disturbance results in shifts from diurnal to nocturnal activity [50–52]. According to our investigation, deer display diurnal behavior with peak activity during crepuscular hours and serow exhibit cathemeral behavior, with the exception of serow in autumn when they are nocturnal. Since our study area experiences minimal human disturbance and given that the survey period was outside the hunting season (which begins on 15 November), it seems unlikely that the activity patterns of either species has been significantly impacted by human intervention.

**Table 2. Parameter estimates (β), standard errors (SE), and p-values from the best model or model-averaged estimates for the single-species, single-season occupancy models for sika deer and Japanese serow in summer and autumn in Shirakawa Village, Gifu Prefecture, Japan (2015–2017).**

| | | Parameter | β | SE | P |
|---|---|---|---|---|---|
| Japanese serow | summer | Ψ(2016) | 0.33 | 0.83 | 0.69 |
| | | Ψ(2017) | 1.33 | 2.00 | 0.51 |
| | | p (Altitude) | −0.37 | 0.16 | <0.05 |
| | autumn | Ψ(2016) | 1.03 | 0.98 | 0.29 |
| | | Ψ(2017) | 2.15 | 1.60 | 0.18 |
| | | Ψ(Altitude) | −0.06 | 0.17 | 0.73 |
| | | p (Altitude) | −0.06 | 0.13 | 0.65 |
| Sika deer | summer | p (2016) | 1.74 | 0.63 | <0.01 |
| | | p (2017) | 3.21 | 0.64 | <0.01 |
| | | p (Altitude) | 0.32 | 0.15 | <0.05 |
| | autumn | Ψ(Altitude) | 0.66 | 0.62 | 0.29 |
| | | p (2016) | 1.80 | 0.49 | <0.01 |
| | | p (2017) | 1.48 | 0.53 | <0.01 |

**Table 3. Top five single-season, two-species occupancy models for sika deer and Japanese serow in summer and autumn in Shirakawa Village, Gifu Prefecture, Japan, 2015–2017. Ψ = occupancy, p/r = detection probability, AICc wgt = Akaike weight. A or a = sika deer, B or b = Japanese serow.**

| Model | | covariates (detection) | ΔAICc | ΔAICc | AICc wgt | no.Par. |
|---|---|---|---|---|---|---|
| Summer | | | | | | |
| ψ$^A$≠ψ$^{BA}$=ψ$^{Ba}$ | pB≠rBA≠rBa | year | 445.92 | 0.00 | 0.60 | 9 |
| ψ$^A$≠ψ$^{BA}$≠ψ$^{Ba}$ | pB≠rBA≠rBa | year | 448.75 | 2.83 | 0.15 | 10 |
| ψ$^A$≠ψ$^{BA}$=ψ$^{Ba}$ | pB=rBA=rBa | year | 448.76 | 2.84 | 0.15 | 7 |
| ψ$^A$≠ψ$^{BA}$=ψ$^{Ba}$ | pB=rBA=rBa | year+altitude | 450.87 | 4.95 | 0.05 | 8 |
| ψ$^A$≠ψ$^{BA}$≠ψ$^{Ba}$ | pB=rBA=rBa | year | 451.53 | 5.61 | 0.04 | 8 |
| Autumn | | | | | | |
| ψ$^A$≠ψ$^{BA}$=ψ$^{Ba}$ | pB=rBA=rBa | year | 411.08 | 0.00 | 0.46 | 7 |
| ψ$^A$≠ψ$^{BA}$≠ψ$^{Ba}$ | pB=rBA=rBa | year | 413.11 | 2.03 | 0.17 | 8 |
| ψ$^A$≠ψ$^{BA}$=ψ$^{Ba}$ | pB=rBA=rBa | year +altitude | 413.69 | 2.61 | 0.12 | 8 |
| ψ$^A$≠ψ$^{BA}$=ψ$^{Ba}$ | pB≠rBA≠rBa | year | 414.05 | 2.97 | 0.10 | 9 |
| ψ$^A$≠ψ$^{BA}$≠ψ$^{Ba}$ | pB≠rBA≠rBa | year | 415.78 | 4.70 | 0.04 | 10 |

Some previous studies have reported both diurnal and nocturnal activities for serow, demonstrating their flexible circadian behavior [24,53]. The closely related Himalayan serow (*C. thar*), exhibits both diurnal and nocturnal activity, particularly in the early morning and from afternoon to night [54], whereas Sumatran serow (*C. sumatraensis*) demonstrates primarily nocturnal behavior [55], and the Chinese serow (*C. milneedwardsii*) exhibits crepuscular activity [56]. Because the presence of predators, interspecific competition, and anthropogenic disturbance vary considerably among these study areas, such local factors may influence the activity patterns of *Capricornis* species.

We found deer to be crepuscular, both in summer and autumn, which is consistent with previous studies in regions with less human disturbance [57,58]. Similarly, red deer (*C. elaphus*), mule deer (*Odocoileus hemionus*), and white-tailed deer (*O. virginianus*) display crepuscular activity patterns [59–61]. Predation risk is a key factor influencing diel activity in many deer species. However, in Japan, large predators have been absent since the extermination of the gray wolf (*Canis lupus*)

Table 4. The parameter estimates (β) and standard error (SE) of the best supported single season, two-species occupancy models for sika deer and Japanese serow in Shirakawa Village, Gifu Prefecture, Japan, 2015-2017. Result in autumn were obtained through model-averaging. A or a = sika deer, B or b = Japanese serow.

| | Summer | | Autumn | |
|---|---|---|---|---|
| | β | SE | β | SE |
| ψA | 0.85 | 0.08 | 0.84 | 0.10 |
| ψBA | 0.78 | 0.09 | 0.65 | 0.11 |
| ψBa | 0.78 | 0.01 | 0.65 | 0.11 |
| pA: 2015 | 0.12 | 0.06 | 0.16 | 0.06 |
| pA: 2016 | 0.28 | 0.09 | 0.42 | 0.09 |
| pA: 2017 | 0.46 | 0.13 | 0.42 | 0.10 |
| pB: 2015 | 0.03 | 0.03 | 0.16 | 0.05 |
| pB: 2016 | 0.08 | 0.06 | 0.43 | 0.08 |
| pB: 2017 | 0.16 | 0.10 | 0.43 | 0.07 |
| rA: 2015 | 0.14 | 0.04 | 0.18 | 0.06 |
| rA: 2016 | 0.30 | 0.06 | 0.45 | 0.07 |
| rA: 2017 | 0.49 | 0.06 | 0.45 | 0.09 |
| rBA: 2015 | 0.23 | 0.08 | 0.16 | 0.05 |
| rBA: 2016 | 0.45 | 0.10 | 0.43 | 0.08 |
| rBA: 2017 | 0.65 | 0.08 | 0.43 | 0.07 |
| rBa: 2015 | 0.29 | 0.07 | 0.16 | 0.05 |
| rBa: 2016 | 0.53 | 0.08 | 0.43 | 0.08 |
| rBa: 2017 | 0.71 | 0.06 | 0.43 | 0.07 |
| SIF | 1.00 | 0.00 | 1.00 | 0.00 |

[62]; thus, human disturbance may have a greater influence on deer behavior. However, since our study area experiences minimal human disturbance, the observed diurnal or crepuscular activity in deer is likely a natural behavioral pattern rather than a response to human influence.

## Spatial distribution and habitat preferences

Based on the occupancy estimates from our single occupancy model, we found both deer and serow widely across all habitat types, indicating a lack of any particular habitat preference. However, previous studies have provided insights into their habitat use, showing that they exhibit preferences for specific forest types and topographic factors. These studies suggest that both serow and deer tend to prefer broad-leaved and mixed forests [25,30,63]. Additionally, serow prefer steep terrain [47,63] and deer prefer grassland [63]. These habitat preferences are likely linked to food resources and safety against predators. The discrepancy between our findings and those of previous studies might arise from factors, such as varying degrees of human disturbance, different vegetation types, or resource availability. In areas with high human disturbance, habitat conditions may change, potentially influencing the behavior and distribution of deer and serow. In contrast, in regions with minimal human impact, their natural habitat preferences may be more clearly expressed. Another factor could be differences in vegetation and food availability. Areas dominated by broad-leaved forest may provide more suitable habitat for serow, whereas areas rich in herbaceous plants may offer more favorable conditions for deer. Our study area is distinctive in that it encompasses a significant portion of broad-leaved forest, with few plantations, and experiences minimal human disturbance, which may affect the habitat preferences of both species.

In addition to occupancy patterns, our models also revealed that elevation had opposite effects on detection probability for deer and serow; detection probability increased with altitude for deer but decreased for serow. While detection

probability is not a direct measure of habitat preference, this contrast may reflect differences in behavior or visibility along elevational gradients. For example, deer may be more active or detectable at higher elevations, while serow may be more frequently encountered at lower elevations. These tendencies, although indirect, may contribute to apparent habitat use patterns and should be considered when interpreting species distributions. Further research is essential to clarify these inconsistencies.

## Implications for competition and coexistence

The recent population growth and range expansion of deer in Honshu may have led to a decline in the serow population or a shift in their habitat use due to vegetation changes associated with deer overpopulation [10,19,64]. In Shirakawa Village, serow were already widely distributed by the 1970s, whereas deer were observed only infrequently until 2010, but since then the species' range has been expanding gradually [29,48].

Despite the potential for competition, we did not observe clear temporal or spatial segregation between serow and deer. The lack of segregation in our study might be related to the relationship between the deer population density and food availability. In general, serow are more selective feeders, focusing primarily on browsing and showing a preference for digestible and nutritional plants [65,66], whereas deer are known to be generalist grazer-browsers, consuming a wide range of plant species including grasses, leaves, shoots, and forbs [13,67]. Since there was no observable vegetation degradation and because food resources are abundant in Shirakawa Village, it appears that serow and deer may be able to coexist, possibly due to differences in their dietary preferences [10,19]. Additionally, while the deer population has increased, it is possible that the density is not yet sufficient to trigger competitive interactions in our study area. Kogane-zawa [21] demonstrated that serow numbers decline in areas where the deer density exceeds 25 individuals/km$^2$, but not where deer density is below 10 individuals/km$^2$. In the case of Mt. Odaigahara, it has been suggested that the impact on natural vegetation is minimal when the deer density is between 5–10 individuals/km² [68]. Considering that we did not observe significant vegetation degradation in our study area, we can infer that the deer density remains below the carrying capacity. Thus, the deer density is not sufficiently high to cause significant spatiotemporal segregation between them and serow.

Our results showed that while the activity times of serow and deer greatly overlapped, there was a slight reduction in overlap and there were distinct peak activity times in autumn. This difference in peak activity times between the species in autumn, may play a role in mitigating competition and potential interactions, corresponding with findings from a previous study [58]. In our study, the RAI values for deer were higher in autumn than in summer, coinciding with a shift in the activity pattern of serow from cathemeral to nocturnal. This increase in RAI values for deer during autumn is likely to be related to their rutting behavior, where males form harems, leading to localized density increases and intensified aggressive interactions. Given that serow are smaller than deer in Honshu [69,70], they might avoid encounters with the more dominant deer, whereas deer tend to be indifferent to the presence of serow [47]. By minimizing any overlap in their peak activity periods, serow may avoid deer to reduce direct competition for resources. In China, for example, Chinese serow (*C. m. milneedwardsii*) were found to exhibit a shift in their activity patterns, being more active during the afternoon and at night in the dry season and between sunrise and noon in the wet season, likely as an adjustment to avoid interference competition with red serow (*C. rubidus*) [71]. Since the probability of occupancy and the RAI index of deer increased during the study period, this could escalate resource and habitat competition between deer and serow in Japan in the future, with potentially increased temporal segregation or the emergence of spatial segregation as a competition mitigation strategy.

## Conclusion

In conclusion, our findings highlight that Japanese serow and sika deer in Shirakawa Village did not exhibit clear temporal or spatial segregation and appear to have the potential to coexist. However, serow altered their circadian activity slightly in autumn, perhaps to reduce resource competition with deer. This suggests that temporal segregation may contribute to

coexistence in areas where the deer population is increasing. However, if the sika deer population continues to increase, competition for habitat and foods may intensify, potentially affecting the behavior and distribution of the Japanese serow. To mitigate these impacts, effective deer population management is necessary, including targeted culling in high-density areas and regular population monitoring. A key aspect of this management is determining the threshold deer density beyond which serow populations experience significant ecological or behavioral shifts. Establishing this threshold will enable the implementation of evidence-based conservation strategies to maintain ecological balance and facilitate the long-term coexistence of these sympatric ungulates.

## Supporting information

**S1 Table. Deer and serow camera trap dataset.**
(CSV)

## Acknowledgments

We express our gratitude to the Shirakawa-Go Wildlife Research Group members, especially T. Sumi, K. Hayashi, and H. Ogawa, for their efforts in collecting field survey data. We also thank R. Kishimoto for invaluable comments on the manuscript. We would also like to thank Dr Mark Brazil, Scientific Editing Services, for assistance in the preparation of the final draft of the manuscript.

## Author contributions

**Conceptualization:** Tomoki Mori, Yasuaki Niizuma.

**Data curation:** Tomoki Mori, Kensuke Miura, Hiroyuki Takeuchi.

**Formal analysis:** Tomoki Mori.

**Investigation:** Tomoki Mori, Kensuke Miura, Hiroyuki Takeuchi.

**Methodology:** Tomoki Mori, Kensuke Miura, Hiroyuki Takeuchi, Yasuaki Niizuma.

**Project administration:** Tomoki Mori, Yasuaki Niizuma.

**Supervision:** Yasuaki Niizuma.

**Visualization:** Tomoki Mori.

**Writing – original draft:** Tomoki Mori.

**Writing – review & editing:** Kensuke Miura, Hiroyuki Takeuchi, Yasuaki Niizuma.

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
