## [Decision Letter · Decision Letter 0]

28 Jan 2025

Dear Dr. Mori,

Thank you for submitting your manuscript to PLOS ONE. After careful consideration, we feel that it has merit but does not fully meet PLOS ONE’s publication criteria as it currently stands. Therefore, we invite you to submit a revised version of the manuscript that addresses the points raised during the review process.

We look forward to receiving your revised manuscript.

Kind regards,

Bogdan Cristescu

Academic Editor

PLOS ONE

Journal Requirements:

2. We note that Figure 1 in your submission contain [map/satellite] images which may be copyrighted. All PLOS content is published under the Creative Commons Attribution License (CC BY 4.0), which means that the manuscript, images, and Supporting Information files will be freely available online, and any third party is permitted to access, download, copy, distribute, and use these materials in any way, even commercially, with proper attribution. For these reasons, we cannot publish previously copyrighted maps or satellite images created using proprietary data, such as Google software (Google Maps, Street View, and Earth). For more information, see our copyright guidelines: http://journals.plos.org/plosone/s/licenses-and-copyright .

We recommend that you contact the original copyright holder with the Content Permission Form (http://journals.plos.org/plosone/s/file?id=7c09/content-permission-form.pdf ) and the following text:

“I request permission for the open-access journal PLOS ONE to publish XXX under the Creative Commons Attribution License (CCAL) CC BY 4.0 (http://creativecommons.org/licenses/by/4.0/ ). Please be aware that this license allows unrestricted use and distribution, even commercially, by third parties. Please reply and provide explicit written permission to publish XXX under a CC BY license and complete the attached form.”

3. Please include captions for your Supporting Information files at the end of your manuscript, and update any in-text citations to match accordingly. Please see our Supporting Information guidelines for more information: http://journals.plos.org/plosone/s/supporting-information .

Additional Editor Comments:

The reviewers have provided a comprehensive roadmap for improving the manuscript. The study is interesting but has a caveat of small sample size which could affect inferences. The implications of pooling data across years in a single season occupancy model with regard to the modelling assumptions must be elaborated. Both these shortcomings must be highlighted in the Discussion, and the Abstract should include a disclaimer on small sample size. Should you decide to revise the manuscript and submit the revision to the journal, please address all comments from the reviewers point by point, either by incorporating in a revised manuscript or by rebuttal-ing the request.

Reviewers' comments:

Reviewer's Responses to Questions

**Comments to the Author**

1. Is the manuscript technically sound, and do the data support the conclusions?

Reviewer #1: Partly

Reviewer #2: Partly

2. Has the statistical analysis been performed appropriately and rigorously?

Reviewer #1: Yes

Reviewer #2: Yes

3. Have the authors made all data underlying the findings in their manuscript fully available?

Reviewer #1: Yes

Reviewer #2: Yes

4. Is the manuscript presented in an intelligible fashion and written in standard English?

Reviewer #1: Yes

Reviewer #2: Yes

Reviewer #1: The authors assess whether the increase in sika deer population may influence the behavior and presence of the Japenese serow. I find this topic very interesting, and it seems timely given the rapid expansion of one species. The manuscript is overall well-structured and provides a basic understanding of the conducted research and the conclusions drawn from it. While I believe the statistical methods are applied appropriately, the underlying assumptions are somewhat problematic. This is primarily due to the lack of information provided in the methods section.

In the discussion the authors mention that the deer population is probably too low to have an effect on serow. This seems to be existing knowledge and not derived from this investigation. This is a key factor and an explanation for many of the discussed aspects earlier on. I suggest to provide this information earlier and explain why the research is still valid and important despite this.

Furthermore, the manuscript contains many grammatical errors and shows signs of insufficient attention to detail. This makes reading the manuscript somewhat cumbersome and I suggest that the authors consult a copyeditor. At this point, I do not believe the manuscript is ready for publication but I would not reject it outright and rather provide the opportunity for revisions.

Abstract

Line 28: Not needed: ...although they coexisted…

Line 31: What are the potential impacts?

Introduction

The introduction is largely well-written and provides a good introduction to the two species and their potential interaction. However, I feel it would benefit from a clearer lead-up to the specific study area and a more explicit explanation of why investigating these two species in this context is important. Are there any broader implications for the ecosystem or for the species themselves? Are there any direct management implications that can be derived from this manuscript? Furthermore, I suggest starting with an explanation or definition of the term sympatric. While it is widely used and most readers are likely familiar with it, the term is central to this paper and should be clarified to avoid any potential confusion. For instance, in lines 57 to 59, are zebras and wildebeests truly sympatric, or are they simply more sympatric compared to other animals in this ecosystem?

I started by pointed out grammatical errors, typos, and formatting issues but I will refrain from doing so throughout the manuscript.

More detailed comments:

Line 59: I recommend double-checking this reference as to my understanding the authors representation do not fully align with the outcomes of Fuller et al. Additionally, this example could be explained somewhat more, similar to the other two given in this context.

Line 69: Selective diet not diets?

Line 69: Inhibits alpine meadows not inhibits in alpine meadows?

Line 73: I believe the species can only be either clearly allopatric or partly sympatric, as these are distinct concepts. Rephrasing this would help improve clarity.

Line 72 - 75: What occurred between 1970 and 1990? Why did the authors choose to report specifically on these two dates?

Line 89 - 91: This sentence needs rephrasing, as it currently implies that the behaviour is solely caused by autumn, whereas the focus is on autumn being the mating season.

Line 80 – 98: This section helps me to understand the relationship between the two species and the significance of autumn, well done. However, in the discussion the authors mention that deer are clearly more dominant than serow. I believe this should be outlined in this section already.

Line 102 - 104: How did it prove ideal for the study species? I suggest eliminating this sentence but if it is important and indeed the ideal choice of method, the authors should explain why.

Materials and methods

I believe the statistical methods are applied correctly and can be used to address the research question. However, I am uncertain whether the underlying assumption of pooling data over multiple years into a single-season model is valid, especially if the locations are not constant. To provide final feedback, I would need more detailed information about the study setup. Please refer to my comments below for further clarification.

Figure 1 should be improved by adding more details, such as clearly marking the borders of the national forest, using more easily distinguishable symbols for the camera traps per year, and especially providing a higher-resolution image. Additionally, the caption should be revised to offer a more thorough and descriptive explanation of the figure. It appears that sampling occurred very close to a human settlement, which contradicts the earlier statement that there was little human disturbance. Furthermore, the figure suggests a spatial separation of locations, while the text states that multiple sites were sampled repeatedly. Please provide clarification.

Line 116: The map does not indicate the boundaries of the national forest, and it would be important to communicate what proportion of the study area is within the national forest. A short explanation what a national forest is would be good too. Furthermore, how do the authors assess the level of human disturbance? Finally and very importantly, what fraction of the survey effort was conducted in each of these areas?

Line 121: Is this relevant given that the study was conducted only between 800 m and 1600 m? Or did I misunderstand something? If it is only for the sake of completeness I suggest adding information about the area above 1600 m too.

Line 118 and line 125: First, it is mentioned that 12% of the area are cedar plantations, but it remains unclear how the other 88% look like. Then, cedar is mentioned again in line 125, is this a repetition or an oversight, or do the authors refer to something else, please explain?

Line 127: How did the authors conclude that there was no degradation of vegetation and food was considered plenty if no data was collected or a reference given? I think some further explanation is needed here.

Line 136: Please explain here that it was only 39 cameras in total over 3 years. This is key information and should be stated clearly.

Line 137: The authors state that cameras were placed in three habitat types to ensure comprehensive representation. However, it seems that the habitat is dominated by a single vegetation type. Based on the provided information, it is difficult to evaluate this clearly, see my previous comments. This lack of information is one of my main concerns with the methods and results of this manuscript.

Line 139: I thought the area primarily consisted of broad-leaved forests but here the authors mention coniferous forest and riparian forest.

Line 140: Does this timeframe ensure demographic closure?

Line 140 – 143: Please provide more details on the spatial placement of these cameras within the study area.

Line 145: Which ones were placed at the same location? Maybe a table would help here.

Line 146: The distance seems to vary greatly please elaborate further on how all of those ensure independence?

Line 166: I believe there is a comma missing after (p), or I do not follow the sentence completely.

Line 167/168: Is this information relevant?

Line 167 – 173: I do not believe this detail is needed, a suitable reference should explain this sufficiently, and a simple sentence would be enough for this manuscript.

Line 174 – 176: This not clear to me. It seems that only a few camera trap locations were sampled repeatedly, so the dataset should be reduced to include only those specific locations? Is it justified to pool these locations, especially if they were not identical, as suggested by the figure? I would appreciate more information that convinces me that this approach is justified. Additionally, the mention of six sampling occasions of 10 days seems odd, considering that the cameras were operating for four months each year. Did the authors reduce the dataset based on the 'early, middle, and late periods of each summer and autumnal month'? This phrasing is unclear to me.

Line 179 – 180: How does an occupancy model determine the habitat characteristics of a species? By providing occupied areas per habitat?

Line 183: I kept wondering about home range sizes for the species, thanks, please give more details.

Line 184: Which assumptions?

Line 185: This clarifies my point from line 179 – 180, please mention this earlier.

Line 186: Covariates not covariances?

Line 196: What justifies keeping detection probability constant?

Line 217: Subordinate because not territorial?

Line 223 - 224: This information should be provided earlier on.

Results

I think the results are well presented and appear logical. However, my concern remains with the modelling assumptions.

Line 250: This sentence could be improved for increased readability.

Table 1: Could this data be presented in a figure? I think this is very important information and a figure is easier to interpret.

Discussion

I suggest structuring the discussion in the same order as the methods and results sections, following the same sequence of topics.

Line 288: I believe it had three vegetation types even though that was never clearly explained and I am not sure about the habitat types.

Line 289: I disagree with the statement that "any potential bias is minimized," as I believe this is an overstatement. The authors seem to acknowledge this by using careful language such as "cautiously interpreted" but I would suggest rephrasing the first part of the statement.

Line 290: Without providing more detail on site selection I do not see how this can be the case.

Line 291: So deer presence should have influenced sika as deer is more dominant? If this is clear than it should be presented earlier.

Line 293-95: Can the authors think of any other explanation for a reduction in the overlap of activity times in autumn that might have to do with food availability or hunting?

Line 299/302: I believe hunting can influence wildlife movement even outside the hunting season. Given the proximity of a village and the apparent presence of hunting, I am not sure it is accurate to say that there is minimal impact. Additionally, the survey stopped just 15 days before the start of the hunting season, and this seems to be an annual occurrence. More information would help to convince the reader that the impact of hunting and human presence, both during and outside the season, has no impact.

Line 312: Now the authors speak of other areas with less human disturbance, while it was stated numerous times already that there is little disturbance in this study system? Please address this inconsistency.

Line 311: Please stick to deer and not sika deer to avoid confusion.

Line 326 – 329: I do not think that listing these options is very convincing, I would prefer to be provided with some fewer options but have those explained more, similar as to what was done in the next sentence.

Line 344 – 346: If this is clear I am wondering if the outcome of this investigation could have not been predicted by this circumstance. In fact if this the case does it not make most other assumptions redundant?

Line 347 – 352: This is very well explained and makes sense to me! As such the inference drawn in line 355 is key in my eyes. Would that not be an explanation for many of the things that have been discussed so far too?

Line 369 – 373: I thought the deer density is not high enough to have an effect?

Reviewer #2: Compliments on an appropriate use of a small data set, to address your hypothesis. Please see my detailed comments below for improving your manuscript -

1) Why is sika considered dominant over serow needs a bit more convincing explanation, the reference provided is based on occupancy, behavioral observation reference would be more convincing to subsequently then support by occupancy B coefficients. The size of Sika in comparison to Serow would be good to mention as well. Early on in the introduction it is mentioned that Serow are territorial while deer are not – is this territoriality behavior limited to conspecifics or extended towards other ungulates as well?

2) The RAI values need to be better used as when reduced to presence non-detection data for occupancy analyses – intensity (relative abundance) information is lost. An RAI bar chart with SE for different vegetation types could be used for depicting habitat use by deer and serow and if availability of vegetation types is available then preference could be computed and compared between deer and serow using indices like Ivlev’s index, etc. adding value to the ecological aspects of the MS.

3) Line 154- please explain what is considered as an independent event

4) Line 186- The covariates for vegetation do not reflect in the results, either remove them or include them as they will provide lot of information regarding habitat use by the two species

5) Line 231- Detected should be replaced by occupy as Psi corrects for detection bias?

6) Line 236 - Independent manner?

7) In the table title replace Japanese deer with sika for consistency

8) Line 261-266 - For both species and both seasons several competing models have substantial AIC support and therefore model averaged Beta coefficients should be the best way to interpret the results of single season occupancy.

9) The Beta coefficients of the best or model averaged for covariates needs to be reported and interpreted in the discussion section. These are important aspects of serow and deer ecology on how vegetation, slope and elevation effect their habitat use.

10) Line 346 - "by partitioning their diets" citation required as no information on diet is present in this MS

11) Line 369 - Local density would increase in pockets due to their rutting behavior, but overall densities should not change unless the study area attracts deer from neighboring areas during rut. Yes aggressive behavior would increase in these pockets.

12) Line 383 - "without pronounced competition" can be omitted as the paper does not provide data to prove if the two species compete or not, only that they coexist without negative negative interactions.

13) Line 385 - "vital for them" can temper this down as your evidence says the contrary (mechanism for coexistence at high deer density.

**Do you want your identity to be public for this peer review?** For information about this choice, including consent withdrawal, please see our Privacy Policy

Reviewer #1: **Yes: ** Tim Hofmann

Reviewer #2: No

---

## [Author Response · Author response to Decision Letter 1]

2 Apr 2025

Additional Editor Comments:

The reviewers have provided a comprehensive roadmap for improving the manuscript. The study is interesting but has a caveat of small sample size which could affect inferences. The implications of pooling data across years in a single season occupancy model with regard to the modelling assumptions must be elaborated. Both these shortcomings must be highlighted in the Discussion, and the Abstract should include a disclaimer on small sample size. Should you decide to revise the manuscript and submit the revision to the journal, please address all comments from the reviewers point by point, either by incorporating in a revised manuscript or by rebutting the request.

Response: Thank you for your comments and for providing a clear roadmap for improving our manuscript. We appreciate the reviewers’ valuable feedback and have carefully addressed each of their concerns. Regarding the small sample size, we have explicitly acknowledged its potential limitations in the Discussion and included a disclaimer in the Abstract, as recommended. We also have elaborated on the implications of pooling data across years in a single-season occupancy model, ensuring that the modeling assumptions are properly discussed. For a detailed account of how each reviewer’s comment was addressed, please refer to our point-by-point responses to the reviewers. Additionally, we have completely revised Figure 1 to comply with the required guidelines. The new figure no longer contains any copyrighted materials and we have updated the figure legend to reflect this revision as follows:

“We created this map ourselves using data provided by the Geospatial Information Authority of Japan (https://www.gsi.go.jp/) and Biodiversity Center of Japan (http://gis.biodic.go.jp/webgis/), under the CC BY 4.0 or a compatible license.”　(Line 147-150)

We appreciate the opportunity to revise our manuscript!

Reviewer #1:

Major comment: The authors assess whether the increase in sika deer population may influence the behavior and presence of the Japanese serow. I find this topic very interesting, and it seems timely given the rapid expansion of one species. The manuscript is overall well-structured and provides a basic understanding of the conducted research and the conclusions drawn from it. While I believe the statistical methods are applied appropriately, the underlying assumptions are somewhat problematic. This is primarily due to the lack of information provided in the methods section.

In the discussion the authors mention that the deer population is probably too low to have an effect on serow. This seems to be existing knowledge and not derived from this investigation. This is a key factor and an explanation for many of the discussed aspects earlier on. I suggest to provide this information earlier and explain why the research is still valid and important despite this.

Response: We sincerely appreciate the reviewer's time and effort in evaluating our manuscript, as well as their positive feedback. We also acknowledge the concerns regarding the underlying assumptions and the need for additional methodological details. In response to these comments, we have revised the Methods section to

provide further clarification on data collection and analytical assumptions.

Additionally, we recognize that the current sika deer population density in our study area may be lower than in other regions where stronger interspecific competition has been observed. As this is a key factor influencing our findings, we have introduced this information earlier in the Introduction, explaining why studying interactions at this stage remains valid and important. Understanding the behavioral responses of Japanese serow under relatively low deer densities provides crucial insights into potential long-term impacts as deer populations continue to expand. In the Discussion, we further emphasize that our findings should be interpreted within the context of deer population density and highlight the importance of ongoing monitoring to assess future competitive interactions as deer densities increase.

Please see more details in our responses to your specific comments below.

Furthermore, the manuscript contains many grammatical errors and shows signs of insufficient attention to detail. This makes reading the manuscript somewhat cumbersome and I suggest that the authors consult a copyeditor. At this point, I do not believe the manuscript is ready for publication but I would not reject it outright and rather provide the opportunity for revisions.

Response: According to your comments, we have consulted a professional English editing service for thorough proofreading and refinement as stated in the acknowledgments.

Abstract

Comment#1 (Line 28): Not needed: ...although they coexisted…

Response: According to your comment, we have removed the phrase "although they coexisted" to improve conciseness as follows:

From: “In Japan, since the 1970s, the endemic Japanese serow (Capricornis crispus) and the sika deer (Cervus nippon) were primarily allopatric, although they coexisted in localized regions.”

To: “In Japan, since the 1970s, the endemic Japanese serow (Capricornis crispus) and the sika deer (Cervus nippon) have been primarily allopatric.”. (Line 26-28)

Comment#2 (Line 31): What are the potential impacts?

Response: We have clarified the potential impacts by specifying concerns about interspecific competition for food and habitat resources, which may influence the distribution and behavior of both species as follows:

From: “This increasing overlap raises concerns about the potential impacts between these two sympatric ungulates.”

To: “The significant and increasing overlap raises concerns about the potential impacts between these two (now sympatric) ungulates, including changes in distribution, shifts in activity patterns, or displacement due to interspecific competition.” (Line 29-32)

Introduction

Comment#4: The introduction is largely well-written and provides a good introduction to the two species and their potential interaction. However, I feel it would benefit from a clearer lead-up to the specific study area and a more explicit explanation of why investigating these two species in this context is important. Are there any broader implications for the ecosystem or for the species themselves? Are there any direct management implications that can be derived from this manuscript? Furthermore, I suggest starting with an explanation or definition of the term sympatric. While it is widely used and most readers are likely familiar with it, the term is central to this paper and should be clarified to avoid any potential confusion. For instance, in lines 57 to 59, are zebras and wildebeests truly sympatric, or are they simply more sympatric compared to other animals in this ecosystem?

I started by pointed out grammatical errors, typos, and formatting issues but I will refrain from doing so throughout the manuscript.

Response: We appreciate the reviewer’s constructive feedback on improving the clarity and structure of the Introduction. According to your comments, we have revised the Introduction to provide a clearer lead-up to the study area, explicitly highlighting why investigating the interactions between these two species in this context is important. We now emphasize the conservation importance of the Japanese serow, including its designation as a "Threatened Local Population" in certain regions, and the management challenges posed by increasing sika deer populations. Additionally, we have clarified the term “sympatric” at the beginning of the Introduction to ensure consistency in its usage throughout the manuscript. We have revised the entire introduction. Please check it.

Added: “Sympatric species are those that coexist in the same geographic area, often sharing similar ecological niches, which creates the potential for competition over limited resources such as habitat and food [1].” (Line 52-54)

More detailed comments:

Comment#5 (Line 59): I recommend double-checking this reference as to my understanding the authors representation do not fully align with the outcomes of Fuller et al. Additionally, this example could be explained somewhat more, similar to the other two given in this context.

Response: According to your comments, we have re-reviewed the reference by Fuller et al. (1989) and found that our previous description did not fully consistent with their findings. We have revised the text to ensure that our representation accurately reflects the study’s outcomes as follows:

From: “In Kenya, three sympatric jackal species exhibit distinct diurnal activity patterns”

To: “In Kenya, three sympatric jackal species partition their activity times, with golden jackal (Canis aureus) being active during the day, while black-backed jackal (Lupulella mesomelas) and side-striped jackal (L. adusta) are mostly nocturnal” (Line 64-66)

Comment#6 (Line 69): Selective diet not diets?

Response: According to your comment, we have revised "selective diets" to "selective diet" to ensure grammatical accuracy as follows:

From: “���, and has a relatively narrow, selective diets”

To: “______, and has a relatively narrow, selective diet” (Line 79)

Comment#7 (Line 69): Inhibits alpine meadows not inhibits in alpine meadows?

Response: According to your comment, we have revised "inhabits in alpine meadows " to "inhabits alpine meadows" to ensure grammatical accuracy as follows:

From: “. It also inhibits in alpine meadows, ______”

To: “It also inhibits alpine meadows” (Line 79-80)

Comment#8 (Line 73): I believe the species can only be either clearly allopatric or partly sympatric, as these are distinct concepts. Rephrasing this would help improve clarity.

Response: According to your comment, we have clarified that the two species were generally allopatric, but there were a few localized exceptions in northern Japan, as follows:

From: “Until the 1970s, these species were clearly allopatric, although deer were also distributed in northern Japan and they partly inhabited sympatrically [14,15]”

To: “Until the 1970s, these species were allopatric, except in a few isolated areas in northern Honshu where their distributions overlapped [15,16]” (Line 83-84)

Comment#9 (Line 72-75): What occurred between 1970 and 1990? Why did the authors choose to report specifically on these two dates?

Response: We selected the period between 1970 and 1990 because it marks a transitional phase in the population dynamics and distribution of sika deer in Japan. Until the 1970s, their populations were relatively stable and confined to certain regions due to historical overhunting and habitat loss. However, since the 1990s, the population has increased significantly, with a dramatic expansion in their range [16], likely due to factors, such as reduced hunting pressure, land-use changes, and milder winters. According to your comments, we have revised the text to clarify this transition and provide context for our choice of these time periods."

From: “However, since the 1990s, the deer population has increased and their distribution (including elevation) has expanded dramatically [16]”

To: “However, since the 1990s the deer population has increased and its distribution (including elevation) has expanded dramatically [17], driven by factors such as reduced hunting pressure, changes in land use, and milder winters [18].” (Line 84-87)

Comment#10 (Line 89-91): This sentence needs rephrasing, as it currently implies that the behaviour is solely caused by autumn, whereas the focus is on autumn being the mating season.

Response: According to your comment, we have revised the sentence to clarify that the increase in local deer density and their heightened activity are due to mating-related behaviors in autumn, rather than the season alone as follows:

From: “Furthermore, the competitive relationship between these two species may vary depending on the season. Since deer in Japan mate in autumn, males congregate at mating sites and establish harems of several females each [16], thereby leading to a temporary rise in their local population density.”

To: “Furthermore, the competitive relationship between these two species may vary depending on seasonal changes in behavior. During the autumn mating season, male deer congregate at mating sites and establish harems of several females each [17], thereby leading to a temporary rise in their local population density.” (Line 109-112)

Comment#11 (Line 80-98): This section helps me to understand the relationship between the two species and the significance of autumn, well done. However, in the discussion the authors mention that deer are clearly more dominant than serow. I believe this should be outlined in this section already.

Response: We agree that outlining the dominance of deer over serow earlier in the manuscript would improve clarity and strengthen the logical flow of the discussion. According to your comment, we have explicitly stated that deer are generally dominant over serow, highlighting their larger body size, aggressive rutting behavior, and ability to outcompete serow for resources as follow:

　　　Added: “Deer can be considered to have a competitive advantage over serow due to their larger body size, social behavior, and broad diet [20–22]. Their ability to exploit a wider range of food resources allows them to outcompete the more specialized serow, particularly in resource-limited habitats [21]. Furthermore, some studies report that serow avoid deer in areas with high deer density [22]. Although serow occasionally display territoriality behaviors toward deer, they rarely succeed in displacing them [23]. Overall, the combination of a larger body size, higher population density, and greater dietary flexibility is thought likely to provide sika deer with a long-term competitive edge in sympatric habitats.”

(Line 91-99)

Comment#12 (Line 102-104): How did it prove ideal for the study species? I suggest eliminating this sentence but if it is important and indeed the ideal choice of method, the authors should explain why.

Response: According to your comment, we have deleted this sentence.

Deleted: “Camera trapping is a widely-used effective and non-invasive tool for studying wildlife [21], which proved ideal for our study area and study species.” (Line 124)

Materials and methods

Comment#13: I believe the statistical methods are applied correctly and can be used to address the research question. However, I am uncertain whether the underlying assumption of pooling data over multiple years into a single-season model is valid, especially if the locations are not constant. To provide final feedback, I would need more detailed information about the study setup. Please refer to my comments below for further clarification.

Response: Thank you for your valuable feedback and for recognizing the appropriateness of our statistical methods in addressing the research question. We appreciate your concern regarding the assumption of pooling data over multiple years into a single-season model, particularly in relation to location consistency.

To address your concern, we provide additional details on our study setup and data collection methodology. We have carefully reviewed your subsequent comments and provided detailed responses to each of them. Please find our point-by-point responses below for your review.

Comment#14: Figure 1 should be improved by adding more details, such as clearly marking the borders of the national forest, using more easily distinguishable symbols for the camera traps per year, and especially providing a higher-resolution image. Additionally, the caption should be revised to offer a more thorough and descriptive explanation of the figure. It appears that sampling occurred very close to a human settlement, which contradicts the earlier statement that there was

---

## [Decision Letter · Decision Letter 1]

3 Jul 2025

Dear Dr. Mori,

Thank you for submitting your manuscript to PLOS ONE. After careful consideration, we feel that it has merit but does not fully meet PLOS ONE’s publication criteria as it currently stands. Therefore, we invite you to submit a revised version of the manuscript that addresses the points raised during the review process.

After R1 the paper is much improved. Reviwer 2 indicate several points to modify such us resolving contradictions in abstract, more details in methodology and some context to tables. After this revision, the paper can be published.

We look forward to receiving your revised manuscript.

Kind regards,

Laurentiu Rozylowicz, Ph.D.

Academic Editor

PLOS ONE

Journal Requirements:

Reviewers' comments:

Reviewer's Responses to Questions

**Comments to the Author**

Reviewer #1: All comments have been addressed

Reviewer #3: All comments have been addressed

2. Is the manuscript technically sound, and do the data support the conclusions?

Reviewer #1: Yes

Reviewer #3: Yes

3. Has the statistical analysis been performed appropriately and rigorously?

Reviewer #1: Yes

Reviewer #3: Yes

4. Have the authors made all data underlying the findings in their manuscript fully available?

Reviewer #1: Yes

Reviewer #3: (No Response)

5. Is the manuscript presented in an intelligible fashion and written in standard English?

Reviewer #1: Yes

Reviewer #3: Yes

Reviewer #1: Congratulations to the authors, I believe this manuscript has greatly improved and I see it fit for publication.

Reviewer #3: This is an interesting and generally well-written paper, particularly with the additions and clarifications made following the first round of review. Below are my general and specific comments.

Abstract:

There is contradiction between the results and the conclusion in the abstract.

line 35 Although

our study was limited by a small sample size, it revealed no clear temporal or spatial

segregation between the species, suggesting that there is potential for coexistence in

shared habitats without pronounced competitive conflict

And

line 43 Although our results are limited, they exhibit a similar

trend to previous studies, suggesting that temporal segregation may play a crucial role in

facilitating the sympatric coexistence of these ungulates.

If one reads the entire manuscript will understand but only from the abstract is confusing.

Introduction

Although at line 71 it is explained that the endemic Japanese

serow (Capricornis crispus) (hereafter serow) and the more wide-ranging sika deer

(Cervus nippon) (hereafter deer), there are several places where sika deer instead of deer appears. Please check carefully all the text.

Line 234 I doubt that the detection is constant. One possibility will be to keep occupancy constat and model detection and then use the best model for detection in modeling the occupancy. In the end there aren’t so many covariates.

Line 241 add a reference for the approach used

Line 297 Not clear, please reformulate. I find the phrasing "tend to be occupied at the same site" a bit awkward.

Results: I am surprised to see that a 90 % CI was used. I think you need to justify the choice given that 95% CI is more often used.

Tabele 3 and Table 4. All the abbreviation should be explained in the table caption. You did this for occupancy and detection but it is needed also for la A, B, a;

Discussion

Line 418-420 the statement need some references

**Do you want your identity to be public for this peer review?** For information about this choice, including consent withdrawal, please see our Privacy Policy

Reviewer #1: **Yes: ** Tim Hofmann

Reviewer #3: No

---

## [Author Response · Author response to Decision Letter 2]

25 Jul 2025

Additional Editor Comments:

Response: Thank you for submitting your manuscript to PLOS ONE. After careful consideration, we feel that it has merit but does not fully meet PLOS ONE’s publication criteria as it currently stands. Therefore, we invite you to submit a revised version of the manuscript that addresses the points raised during the review process.

After R1 the paper is much improved. Reviwer 2 indicate several points to modify such us resolving contradictions in abstract, more details in methodology and some context to tables. After this revision, the paper can be published.

Response: Thank you very much for the opportunity to revise our manuscript. We appreciate the positive feedback from the editor and reviewer. In response to Reviewer 2’s suggestions, we have carefully resolved the contradictions in the abstract, provided additional methodological details, and added necessary context and explanations to the tables. We believe these improvements have addressed the reviewer's concerns and enhanced the clarity and readability of our manuscript.

Thank you again for your valuable feedback and consideration.

Reviewer #2:

Response: We sincerely appreciate the reviewer's time and effort in evaluating our manuscript, as well as their positive feedback!

<Abstract>

Comment#1: There is contradiction between the results and the conclusion in the abstract.

line 35 Although our study was limited by a small sample size, it revealed no clear temporal or spatial segregation between the species, suggesting that there is potential for coexistence in shared habitats without pronounced competitive conflict

and

line 43 Although our results are limited, they exhibit a similar trend to previous studies, suggesting that temporal segregation may play a crucial role in facilitating the sympatric coexistence of these ungulates.

If one reads the entire manuscript will understand but only from the abstract is confusing.

Response: Thank you for the comment. We acknowledge the confusion caused by the contradictory statements in the abstract. According to your suggestion, we have revised line 43-44 of the abstract as follows:

From: “Although our results are limited, they exhibit a similar trend to previous studies, suggesting that temporal segregation may play a crucial role in facilitating the sympatric coexistence of these ungulates.”

To: “This seasonal adjustment indicates a context-dependent behavioural response that may serve to reduce temporal overlap and mitigate competition”

<Introduction>

Comment#2: Although at line 71 it is explained that the endemic Japanese serow (Capricornis crispus) (hereafter serow) and the more wide-ranging sika deer (Cervus nippon) (hereafter deer), there are several places where sika deer instead of deer appears. Please check carefully all the text.

Response: We have carefully reviewed the entire manuscript and unified the terminology by consistently using "serow" and "deer" to refer to sika deer, as defined in line 71. However, we have retained the full species name "sika deer" in tables and figure captions for clarity and readability.

Comment#3 (Line 234): I doubt that the detection is constant. One possibility will be to keep occupancy constat and model detection and then use the best model for detection in modeling the occupancy. In the end there aren’t so many covariates.

Comment#6 (Results): I am surprised to see that a 90 % CI was used. I think you need to justify the choice given that 95% CI is more often used.

Response: Thank you for your valuable comments. According to your comments, we have conducted model selection for both occupancy and detection probabilities under the constraint of including at most one categorical and one continuous variable for each component, taking into account the limited sample size (n = 39). In other words, each model included no more than a total of four covariates across occupancy and detection. Additionally, due to technical limitations, we reanalyzed the single-species models using the R package unmarked instead of PRESENCE.

Furthermore, we revised our inference approach by replacing the previously used 90% confidence intervals with standard p-values based on model-averaged estimates.

As a result of incorporating covariates into the detection component, some changes were observed in detection-related results. However, these changes did not affect the main conclusions of the study.

Please refer to the revised Methods and Results sections for details.

Comment#4 (Line 241): add a reference for the approach used

Response: According to your comment, we have add the following reference.

Fuller, A. K., D. W. Linden, and J. A. Royle. 2016. Management decision making for fisher populations informed by occupancy modeling. Journal of Wildlife Management 80:794–802.

Comment#5 (Line 297): Not clear, please reformulate. I find the phrasing "tend to be occupied at the same site" a bit awkward.

Response: According to your comment, we have revised the sentence at line 297 to improve clarity as follows:

From: “The estimate of SIF(φ) < 1 is interpreted as indicating that the two species tend to be occupied at the same site less frequently than expected under the assumption that the species select space independently (i.e., spatial segregation). The estimate of SIF(φ)>1 is interpreted as indicating that the two species tend to be occupied at the same site more frequently than expected under the assumption that the species select space independently (i.e., spatial overlap). A SIF (φ) value of 1 indicates that the two species use a site in an independent manner.”

To: “The estimate of SIF (φ) < 1 suggests that the two species co-occur less frequently than expected under the assumption of independence (i.e., spatial segregation). Conversely, SIF (φ) > 1 indicates that the two species co-occur more frequently than expected under independence (i.e., spatial overlap). A value of SIF (φ) = 1 implies that the two species occupy sites independently.”

Comment#7 (Table 3,4): All the abbreviation should be explained in the table caption. You did this for occupancy and detection but it is needed also for la A, B, a;

Response: We appreciate the reviewer's suggestion. However, the subscripts such as "a" and "b" in parameters (e.g., pA, pB, ψBA, ψBa) represent conditional states of occupancy and detection probabilities and have already been clearly defined in the Methods section following the standard parameterization by MacKenzie et al. (2004, 2006). Repeating these definitions in the table caption would introduce redundancy and could lead to confusion by unnecessarily increasing complexity. To maintain clarity and readability, we explicitly define only the species abbreviations (A = sika deer; B = Japanese serow) in the caption, as these directly pertain to species identities and are essential for immediate interpretation.

Comment#8 (Line 418-420): the statement need some references

Response: We have revised the wording to improve clarity. While we did not add new references, the sentence was rephrased to present the statement more cautiously and avoid the need for citation.

From: “The discrepancy between our findings and those of previous 439 studies might arise from factors,

such as varying degrees of human disturbance, different vegetation types, or resource availability.”

To: “Because the presence of predators, interspecific competition, and anthropogenic disturbance vary considerably among these study areas, such local factors may influence the activity patterns of Capricornis species.”

---

## [Editor Report · Decision Letter 2]

29 Jul 2025

Temporal and spatial interactions in sympatric ungulates: insights from Japanese serow and sika deer

PONE-D-24-46621R2

Dear Dr. Mori,

We’re pleased to inform you that your manuscript has been judged scientifically suitable for publication and will be formally accepted for publication once it meets all outstanding technical requirements.

Kind regards,

Laurentiu Rozylowicz, Ph.D.

Academic Editor

PLOS ONE
---

## [Editor Report · Acceptance letter]

PONE-D-24-46621R2

PLOS ONE

Dear Dr. Mori,

I'm pleased to inform you that your manuscript has been deemed suitable for publication in PLOS ONE. Congratulations! Your manuscript is now being handed over to our production team.

Kind regards,

on behalf of

Dr. Laurentiu Rozylowicz

Academic Editor

PLOS ONE